# Direct Density Ratio Optimization:
# A Statistically Consistent Approach to Aligning Large Language Models

**Rei Higuchi** [1] [2]   **Taiji Suzuki** [1] [2]

## Abstract

Aligning large language models (LLMs) with human preferences is crucial for safe deployment, yet existing methods assume specific preference models like Bradley-Terry model. This assumption leads to statistical inconsistency, where more data doesn't guarantee convergence to true human preferences. To address this critical gap, we introduce a novel alignment method Direct Density Ratio Optimization (DDRO). DDRO directly estimates the density ratio between preferred and unpreferred output distributions, circumventing the need for explicit human preference modeling. We theoretically prove that DDRO is statistically consistent, ensuring convergence to the true preferred distribution as the data size grows, regardless of the underlying preference structure. Experiments demonstrate that DDRO achieves superior performance compared to existing methods on many major benchmarks. DDRO unlocks the potential for truly data-driven alignment, paving the way for more reliable and human-aligned LLMs.

## 1. Introduction

Large language models (LLMs) have achieved remarkable performance across various natural language processing tasks, including question answering, text generation, and translation (Achiam et al., 2023; Chowdhery et al., 2023; Touvron et al., 2023). However, they can also generate outputs containing harmful information, biased opinions, and misinformation, posing potential risks to society (Bender et al., 2021; Weidinger et al., 2021; Bai et al., 2022). To develop safe and reliable LLMs, it is essential to align their behavior with human intentions, values, and ethics (Christiano et al., 2017; Gabriel, 2020; Glaese et al., 2022).

[1]University of Tokyo, Tokyo, Japan [2]Center for Advanced Intelligence Project, RIKEN, Tokyo, Japan. Correspondence to: Rei Higuchi <higuchi-rei714@g.ecc.u-tokyo.ac.jp>.

*Proceedings of the 42$^{nd}$ International Conference on Machine Learning*, Vancouver, Canada. PMLR 267, 2025. Copyright 2025 by the author(s).

Recently, alignment methods for LLMs, such as Reinforcement Learning from Human Feedback (RLHF) (Ouyang et al., 2022; Christiano et al., 2017; Stiennon et al., 2020) and Direct Preference Optimization (DPO) (Rafailov et al., 2024), have yielded significant progress. RLHF learns a reward model based on human feedback and optimizes the LLM's policy via reinforcement learning to maximize this reward. DPO directly optimizes the LLM's policy from human preference data without explicitly learning a reward model, and it has been reported to be more computationally efficient and stable than RLHF.

However, these existing methods rely on the assumption that human preferences follow a specific model (e.g., the Bradley-Terry model (Bradley & Terry, 1952)). This assumption may not accurately capture the complexity of real human preferences (Wu et al., 2022), leading to a fundamental issue of a lack of statistical consistency. Statistical consistency refers to the property that, as the amount of data increases, the learned model converges to the true optimal model (Vapnik, 1998; Mohri et al., 2012). Without statistical consistency, increasing the amount of data does not guarantee performance improvement, raising concerns about the model's reliability and safety. More concretely, even if one perfectly optimizes the loss function based on the Bradley-Terry model, it does not necessarily guarantee the accurate learning of the true underlying human preferences.

In this work, we aim to address the statistical consistency problem in existing methods and develop a more theoretically justified alignment method that benefits from increased data. Specifically, we propose a novel alignment method, "Direct Density Ratio Optimization (DDRO)," which directly estimates the distribution of preferable outputs (Sugiyama et al., 2012a; 2007b; Kanamori et al., 2009) without relying on human preference models. DDRO takes an approach that directly estimates the density ratio between the distributions of preferable and unpreferable outputs. This allows for direct alignment from data without depending on certain human preference models. Our contributions can be summarized as follows:

1. Methodological Contribution: We propose a new preference optimization method called DDRO that does not rely on any human preference model. The idea of our

proposal is to frame the preference optimization problem into a *direct density ratio estimation* problem between the preferred and unpreferred data distributions, and solve it by matching the true ratio and our estimator's ratio through minimizing their *Bregman divergence*. DDRO does not require paired data, which contrasts with such a method as DPO that requires paired data. We also discuss how our method is connected to existing work by investigating how the preferred and unpreferred data distributions interact in the loss function.

2. Theoretical Contribution: We theoretically prove that DDRO is statistically consistent, ensuring convergence to the true distribution as the number of data points increases, without dependence on any preference model. This stems from the usage of the Bregman divergence for density ratio estimation because it gives an upperbound of the $L^2$-distance from the true distribution of the preferred data.

3. Practical Contribution: We validate DDRO across multiple real-world benchmarks, showing that it achieves performance on par with or surpassing widely-adopted methods such as KTO and BCO. Even when converting paired datasets into an unpaired format, DDRO nearly matches or outperforms DPO, which is a paired method.

The remainder of this paper is organized as follows. Section 2 provides background knowledge on LLMs and alignment, as well as existing alignment methods and their limitations. Section 3 describes the detailed theoretical framework of our proposed method, Direct Density Ratio Optimization (DDRO), including its simplified version, and discusses its statistical consistency. Section 4 presents the experimental setup and results to validate the effectiveness of our proposed method. Section 5 discusses the experimental results, the strengths and weaknesses of our method, and the significance and contributions of this research. Finally, Section 6 concludes this paper and outlines future directions.

## 2. Related Work: Limitations of Existing Alignment Methods

Large language models (LLMs) have achieved remarkable progress in natural language processing. However, they also present risks, notably the generation of harmful or inappropriate content. Aligning LLMs with human values and preferences is therefore a critical challenge (Christiano et al., 2017; Gabriel, 2020; Glaese et al., 2022). Reinforcement Learning from Human Feedback (RLHF) (Ouyang et al., 2022; Christiano et al., 2017; Stiennon et al., 2020) has emerged as a promising paradigm for this alignment, demonstrating effectiveness in enhancing helpfulness, harmlessness, and alignment with complex instructions (Zheng et al., 2024; Dai et al., 2023).

RLHF consists of a two-stage process. The first stage involves learning a reward model $r_\phi(x, y)$ that predicts the quality of a response $y$ to a given prompt $x$, based on human preference data. This is achieved using a dataset of human comparisons $\mathcal{D}_{\text{paired}} = \{(x_i, y_i^+, y_i^-)\}_{i=1}^N$, where each data point $(x_i, y_i^+, y_i^-)$ indicates that human annotators prefer response $y_i^+$ over $y_i^-$ for prompt $x_i$. The objective of reward model learning is to acquire a reward function that is consistent with human preferences.

In RLHF, the learning of the reward model employs a loss function $\mathcal{L}_{\text{RM}}(\phi)$ based on the Bradley-Terry model. The Bradley-Terry model is used to model human preference probabilities. For responses $y_1, y_2$ to a prompt $x$, the probability $\Pr(y_1 \succ y_2 | x)$ that response $y_1$ is preferred over $y_2$ is defined as

$$\Pr(y_1 \succ y_2 | x) = \sigma(r_\phi(x, y_1) - r_\phi(x, y_2))$$

where $\sigma(z) = (1 + e^{-z})^{-1}$ is the sigmoid function.

Based on this model, the loss function $\mathcal{L}_{\text{RM}}(\phi)$ is formulated to minimize the expected binary cross-entropy loss in predicting human preferences in the paired dataset $\mathcal{D}_{\text{paired}}$:

$$\mathcal{L}_{\text{RM}}(\phi) = -\mathbb{E}_{\mathcal{D}_{\text{paired}}} \left[ \log \sigma(r_\phi(x, y^+) - r_\phi(x, y^-)) \right]$$

In the second stage, the learned reward model is used to optimize the LLM policy $p_\theta(y \mid x)$ using reinforcement learning algorithms. This is typically done using Proximal Policy Optimization (PPO) (Schulman et al., 2017) by maximizing the following objective function:

$$\max_\theta \mathbb{E}_{x \sim \mathcal{D}, y \sim p_\theta(\cdot|x)}[r_\phi(x, y)] - \beta \text{KL}[p_\theta(y \mid x) \| p_{\text{ref}}(y \mid x)], \tag{1}$$

Here, $p_{\text{ref}}(y \mid x)$ is the reference policy (often the initial LLM policy), and $\beta$ is a hyperparameter that controls the strength of KL divergence regularization. The KL divergence term contributes to stabilizing training and mitigating reward hacking (Ziegler et al., 2019).

While RLHF has been empirically successful, it has limitations such as high computational costs due to separate reward model and policy learning, and instability inherent in reinforcement learning (Rafailov et al., 2024). To address these issues, Direct Preference Optimization (DPO) (Rafailov et al., 2024) was proposed. DPO aims to streamline the alignment process by directly optimizing the policy from preference data, bypassing explicit reward modeling. DPO leverages the theoretical connection between reward functions and optimal policies, demonstrating improved stability and computational efficiency compared to RLHF. However, DPO inherits a critical assumption from the underlying Bradley-Terry model: that human preferences can be accurately modeled by this specific parametric form. This reliance on the Bradley-Terry model raises concerns about statistical consistency. Specifically, even DPO converges to

the optimal policy under the Bradley-Terry model assumption as paired comparison data increases, convergence to the true optimal policy reflecting actual human preferences is not guaranteed if human preferences deviate from this model.

**Proposition 2.1.** *There exists a class of preferences, none of which can be obtained by minimizing the negative log-likelihood loss under the Bradley-Terry model assumption.*

See Appendix A for the proof. Proposition 2.1 suggests that the performance of DPO may be fundamentally limited by the validity of the Bradley-Terry model assumption. Furthermore, both RLHF and DPO rely on paired comparison data. Collecting such data at scale can be expensive and time-consuming, potentially hindering the scalability of these alignment approaches.

To mitigate the reliance on paired comparison data and potentially improve scalability, Kahneman-Tversky Optimization (KTO) (Ethayarajh et al., 2024) introduced the use of unpaired preference data. Unpaired data, where individual responses are labeled with scores or binary feedback (e.g., "good" or "bad"), is often more readily available and scalable as it can be collected from real-world user interactions. KTO aligns LLMs using such unpaired data by leveraging prospect theory (Kahneman & Tversky, 2013; Tversky & Kahneman, 1992), which captures the asymmetry in human value functions, particularly the higher sensitivity to losses compared to gains. While KTO offers a practical approach by utilizing more accessible unpaired data, it also assumes a preference model based on prospect theory. Similar to DPO's reliance on the Bradley-Terry model, the performance and statistical consistency of KTO are contingent on the validity of prospect theory assumption in capturing human preferences.

In contrast to DPO and KTO, which depend on specific parametric models for human preferences, our proposed approach aims to achieve statistical consistency without assuming a particular preference model. While we also utilize unpaired preference data as in KTO, the key novelty of our approach lies in its ability to learn a policy that is guaranteed to converge to the distribution of preferred outputs as the amount of unpaired data increases, regardless of whether human preferences strictly adhere to Bradley-Terry, prospect theory, or other predefined models. Such statistical consistency is crucial for ensuring the reliability and robustness of LLM alignment. Especially in real-world scenarios where human preferences are complex and diverse, and may be poorly modeled by simplistic assumptions, our approach offers a significant advantage.

# 3. Direct Density Ratio Optimization (DDRO)

To avoid relying on a specific preference model such as Bradley-Terry model, we propose a new method called *Direct Density Ratio Optimization* (DDRO) that estimates the aligned model through density ratio estimation. The basic idea of existing preference optimization methods (DPO, PPO, KTO) is to estimate the density ratio between the aligned model and the given reference model. Indeed, the optimal solution for PPO (1) can be given as $p^* \propto \exp(-r_\phi/\beta)p_{\text{ref}}$, that is, PPO obtains the solution $p^*$ via the *density ratio* $p^*/p_{\text{ref}}$ which is proportional to $\exp(-r_\phi/\beta)$. The same argument applies to DPO and KTO. Based on this observation, our idea is to directly estimate the density ratio without going through any intermediate step such as estimating an artificial preference model (like Bradley-Terry model) or constructing a reward model. For that purpose, we utilized a technique developed in the field of direct density ratio estimation (Sugiyama et al., 2012b). To introduce our method, we begin by describing the problem setting. We then derive the DDRO objective through *Bregman divergence*, explain its advantage from a theoretical viewpoint, and conclude by providing a practical, simplified variant of the approach.

## 3.1. Problem Setting

The alignment of large language models can be formulated as an optimization problem aimed at bringing the model's output distribution closer to a desirable human preference distribution. Specifically, given a prompt $x$ from a prompt set $\mathcal{X}$, our goal is to align the conditional distribution of responses $y$ denoted as $p_\theta(y \mid x)$ with the unknown true preferred response distribution $p^+(y \mid x)$. This alignment is achieved through density ratio estimation in our method. We also denote $p_{\text{x}}(x)$ as the probability distribution of a prompt $x \in \mathcal{X}$. Here, we consider responses $y$ to be elements of a response set $\mathcal{Y}$.

**Assumption 3.1** (Assumption on Response Set). We assume that the prompt set $\mathcal{X}$ and response set $\mathcal{Y}$ are finite.

This is a practically motivated assumption. For both prompts and responses, large language models operate on sequences of tokens from a finite vocabulary $\mathcal{V}$. Given the context window length $C$ and generation length limit $T$, the size of possible sequences is bounded by $|\mathcal{V}|^C$ for $\mathcal{X}$ and $|\mathcal{V}|^T$ for $\mathcal{Y}$, ensuring both sets are finite. Technically, this assumption is particularly required to ensure that we can calculate the density function (or more precisely, the probability mass function). As long as we can obtain a density or mass function, then we may remove this condition.

Our goal is to align this model distribution with a true preferred distribution $p^+(y \mid x)$, which represents the responses that humans find desirable. We also denote

$p^-(y \mid x)$ as the corresponding true unpreferred response distribution. Although $p^+(y \mid x)$ is unknown, we assume access to unpaired preference data $\mathcal{D}_+ = (x_i^+, y_i^+)_{i=1}^{n^+}$ and $\mathcal{D}_- = (x_i^-, y_i^-)_{i=1}^{n^-}$ sampled from $p^+$ and $p^-$ respectively, where $n^+ = |\mathcal{D}_+|$ and $n^- = |\mathcal{D}_-|$. We remark that our method requires only unpaired data and does not require a human preference modeling that describes the relation between pairs. This property is more like KTO rather than DPO which requires paired data.

Here we consider a situation where the data are generated through the reference model $p_{\mathrm{ref}}$ and human taggers give labels of "preferred" and "unpreferred." Then, we have the following relation between $p_{\mathrm{ref}}$, $p^+$ and $p^-$:

$$p_{\mathrm{ref}}(y \mid x) = p^+(y \mid x)p(+ \mid x) + p^-(y \mid x)p(- \mid x), \quad (2)$$

where $p(+ \mid x)$ and $p(- \mid x)$ are the probability of preferred and unpreferred labels, respectively. As we have described above, the preference optimization can be reduced to a problem of estimating the density ratio $r^*(y|x) = p_{\mathrm{ref}}(y|x)/p^+(y|x)$. Indeed, if we know $r^*(y|x)$, then we can recover $p^+(y|x)$ as $p^+(y|x) = p_{\mathrm{ref}}(y|x)/r^*(y|x)$. From Eq. (2), the density ratio can be rewritten as $r^*(y \mid x) = p(+ \mid x) + p(- \mid x)\frac{p^-(y|x)}{p^+(y|x)}$, which means that estimating $r^*$ is reduced to estimating another density ratio $\frac{p^-(y|x)}{p^+(y|x)}$. Hence, DDRO aims to estimate this density ratio $g^*(y \mid x) = \frac{p^-(y|x)}{p^+(y|x)}$ to obtain $r^*$ using $\mathcal{D}_+$ and $\mathcal{D}_-$. Here, we place the assumption that $p(+ \mid x)$ is a constant, say $t \in (0,1)$, for all input $x$.

**Assumption 3.2** (Assumption on Reference Distribution). The reference distribution $p_{\mathrm{ref}}$ can be expressed as a convex combination of $p^+$ and $p^-$:

$$p_{\mathrm{ref}} = tp^+ + (1-t)p^-, \quad t \in (0,1).$$

In other words, the probability of preferred data is constant: $p(+ \mid x) = t \ (\forall x \in \mathcal{X})$.

Under this condition, we can see $g^*$ and $r^*$ satisfy the following relationship:

$$g^* = \frac{1}{1-t}r^* - \frac{t}{1-t}. \quad (3)$$

Therefore, the true density ratio $r^*$ can be estimated by estimating $g^*$ from data. In the next section, we derive a *Bregman divergence* loss to estimate $g^*$, which is the key component of our proposal.

### 3.2. DDRO: The Proposed Method

Following the argument described above, DDRO directly matches the ratio between $p_\theta$ and $p_{\mathrm{ref}}$ to the true ratio $g^*$ by minimizing the Bregman divergence loss: Let $f : \mathbb{R} \to \mathbb{R}$ be a differentiable strictly convex function with $f(1) = 0$, then we define the Bregman divergence loss for the density ratio estimator as

$$\mathcal{L}_{\mathrm{Breg}}(\theta) = \mathbb{E}_{x \sim p_x, y|x \sim p^+} \left[ \mathrm{Breg}_f\big(g_\theta \,\|\, g^*\big) \right],$$

where

$$\mathrm{Breg}_f(g\|\tilde{g}) := f(\tilde{g}) - f(g) - f'(g)(\tilde{g} - g),$$
$$g_\theta := \frac{1}{1-t}\frac{p_{\mathrm{ref}}}{p_\theta} - \frac{t}{1-t}.$$

From the convexity of $f$, $\mathcal{L}_{\mathrm{Breg}}(\theta)$ is always non-negative and equals to 0 if and only if $g_\theta = g^*$, $(p^+ \times p_x)$-almost surely. Moreover, from Eq. (3), we can also notice that $p_\theta = p^+$ if and only if $g_\theta = g^*$. This observation ensures that we may estimate the preferred data distribution $g^+$ by minimizing the Bregman divergence loss $\mathcal{L}_{\mathrm{Breg}}(\theta)$. Now, we also define $\tilde{p}_\theta = \frac{1}{1-t}p_{\mathrm{ref}} - \frac{t}{1-t}p_\theta$, which corresponds to an estimator of the unpreferred data distribution $p^-$. Then, we have the following statement.

**Proposition 3.3.** *Suppose that $\hat{\theta}$ minimizes the Bregman divergence loss so that $\mathcal{L}_{\mathrm{Breg}}(\hat{\theta}) = 0$, then it holds that*

$$p_{\hat{\theta}}(y|x) = p^+(y|x), \ \tilde{p}_{\hat{\theta}}(y|x) = p^-(y|x) \ ((p^+ \times p_x)\text{-}a.s.).$$

This motivates us to use the Bregman divergence loss for the preference optimization.

**Empirical estimate of the Bregman divergence loss.** Although we know that minimizing the Bregman divergence loss leads to estimating the preferred data distribution $p^+$, it is not obvious how to calculate the loss from the observed data. However, expanding the definition of the Bregman divergence, we see that

$$\int f(g^*) - f(g_\theta) - f'(g_\theta)(g^* - g_\theta)\mathrm{d}p_+(y|x)$$
$$= \int -f(g_\theta) - f'(g_\theta)\left(\frac{p_-}{p_+} - g_\theta\right)\mathrm{d}p_+(y|x) + (\mathrm{const.})$$
$$= \mathbb{E}_{p_+}\left[-f(g_\theta) + f'(g_\theta)g_\theta\right] - \mathbb{E}_{p_-}\left[f'(g_\theta)\right] + (\mathrm{const.}),$$

where $\mathbb{E}_{p_\pm}[\cdot]$ denotes expectation with respect to $p_\pm(y|x)p_x(x)$. Therefore, the Bregman divergence is represented by the expectations with respect to the preferred and unpreferred data distributions, which can be well approximated by the empirical average over the fine-tuning training datasets: The empirical risk $\hat{\mathcal{L}}_{\mathrm{Breg}}(\theta)$ is given by

$$\hat{\mathcal{L}}_{\mathrm{Breg}}(\theta) = \frac{1}{n^+} \sum_{z_i^+ \in \mathcal{D}_+} \left(-f(g_\theta(z_i^+)) + f'(g_\theta(z_i^+))g_\theta(z_i^+)\right)$$
$$- \frac{1}{n^-} \sum_{z_i^- \in \mathcal{D}_-} \left(f'(g_\theta(z_i^-))\right),$$

where we abbreviated the notation as $g_\theta(z_i) = g_\theta(y_i \mid x_i)$ for $z_i = (x_i, y_i)$. This enables us to directly estimate the density ratio with the consistency guarantee (Proposition 3.3).

Since just minimizing the empirical risk $\hat{\mathcal{L}}_{\text{Breg}}(\theta)$ will lead to catastrophic forgetting, we include a Kullback–Leibler divergence (KL-divergence) regularization in the objective:

$$\mathcal{L}_{\text{DDRO}}(\theta) := \mathcal{L}_{\text{Breg}}(\theta) + \gamma \, \text{KL}(p_\theta \,\|\, p_{\text{ref}}), \qquad (4)$$

where $\text{KL}(p \,\|\, q) := \mathbb{E}_q[\log(p/q)]$. This is the population version of our DDRO objective.

Analogously, its empirical version is also defined as

$$
\begin{aligned}
\hat{\mathcal{L}}_{\text{DDRO}}(\theta) = \\
\frac{1}{n^+} \sum_{z_i^+ \in \mathcal{D}_+} \big(-f(g_\theta(z_i^+)) + f'(g_\theta(z_i^+))g_\theta(z_i^+) \\
+ \gamma t g_\theta(z_i^+)^{-1} \log g_\theta(z_i^+)\big) \\
- \frac{1}{n^-} \sum_{z_i^- \in \mathcal{D}_-} \big(f'(g_\theta(z_i^-)) \\
+ \gamma(1-t)g_\theta^{-1}(z_i^-) \log g_\theta(z_i^-)\big),
\end{aligned} \qquad (5)
$$

where the KL-divergence is also replaced by its empirical estimate based on Assumption 3.2 to represent $p_{\text{ref}}$. We can see that various types of preference optimization methods can be derived from our DDRO formulation (5) by examining various convex functions $f$ for the Bregman divergence (see Appendix F for an empirical investigation of this choice). This offers modeling flexibility depending on the data property.

*Example* 3.4. Here, we give one particular example of the Bregman divergence. Suppose that we employ $f(x) = x \log x - (1 + x) \log(1 + x)$. Then the Bregman loss is reduced to the logistic loss:

$$\mathcal{L}_{\text{Breg}}(\theta) = \mathbb{E}_{p^+} [\log(1 + g_\theta)] + \mathbb{E}_{p^-} [\log(1 + g_\theta^{-1})].$$

Indeed, if we let a "classifier" $f_\theta$ be $f_\theta = \log(1/g_\theta)$, then the loss becomes $\mathcal{L}_{\text{Breg}}(\theta) = \mathbb{E}_{p^+} [\log(1 + \exp(-f_\theta)] + \mathbb{E}_{p^-} [\log(1 + \exp(f_\theta))]$, which coincides with the loss of the logistic regression. We employ this logistic loss for our implementation. See Sugiyama et al. (2012a) for more choices of density ratio estimators.

**Related work on density ratio estimation**    Direct density ratio estimation has been extensively studied in the literature (Sugiyama et al., 2012b). The idea behind this technique is to directly model the ratio between two densities $p(x)/q(x)$ is more efficient and robust, especially in high-dimensional spaces. Density ratio estimation has been applied to various machine learning tasks, including covariate shift adaptation (Shimodaira, 2000; Sugiyama et al., 2007a), mutual

information approximation (Suzuki et al., 2008; 2009), and causal inference (Yamada & Sugiyama, 2010).

The Bregman divergence framework that we employed for DDRO was originally established in (Sugiyama et al., 2012b) and is one of the most general framework for density ratio estimation. This framework encompasses various density ratio estimation methods depending on the choice of Bregman divergence (i.e., the convex function $f$) and how the density ratio model is parameterized. Notable examples include Least Squares Importance Fitting (LSIF) (Kanamori et al., 2009), Kernel Mean Matching (KMM) (Gretton et al., 2009), and Kullback-Leibler Importance Estimation Procedure (KLIEP) (Sugiyama et al., 2007b). Basically, we may employ any density ratio estimator for the preference optimization. However, we employ the logistic loss for our implementation due to its simplicity.

### 3.3. Practical modification of DDRO

While the DDRO is effective in terms of consistency as shown in Theorem 3.3 and Theorem 4.1 below, we found that its training becomes unstable in preliminary experiments due to the appearance of gradient spikes during training (see Appendix E). To mitigate this instability, we propose a practical variant of the DDRO loss that introduces a monotonically increasing function $S$ in the loss of each data point:

$$
\begin{aligned}
\hat{\mathcal{L}}_{\text{DDRO}}(\theta) = \frac{1}{n^+} \sum_{z_i^+ \in \mathcal{D}_+} \big[S\big(\log(1 + g_\theta(z_i^+))\big) \\
+ \gamma t g_\theta(z_i^+)^{-1} \log g_\theta(z_i^+)\big] \\
+ \frac{1}{n^-} \sum_{z_i^- \in \mathcal{D}_-} \big[S\big(\log(1 + g_\theta^{-1}(z_i^+))\big) \\
+ \gamma(1-t)g_\theta^{-1}(z_i^-) \log g_\theta(z_i^-)\big].
\end{aligned} \qquad (6)
$$

We employed $S(x) = \log(\sigma(x))$ for the sigmoid function $\sigma$ in our experiments, which showed good empirical results. We note that we may choose other possibilities for $S$. We provide a comparison of different $S$ in Appendix E. The rationale behind the choice of log-sigmoid function is as follows: Applying $S$ clips large values of $g_\theta$ and its gradient, which prevents gradient spikes and also works as regularization, so that $p_\theta$ does not go far away from the original reference model $p_{\text{ref}}$. Indeed, we observed that this modification can mitigate gradient spikes (see Appendix E).

## 4. Theoretical Analysis

In this section, we give some theoretical analysis of our proposed method. First, we give the error bound of the estimated distribution via the density ratio estimator, which gives justification to use the Bregman divergence loss. Next, we investigate a connection to existing approaches by study-

*Table 1.* Unified perspective on preference optimization objectives. Existing methods apply (partially) increasing and (partially) convex functions such as sigmoid $\sigma(\cdot)$, quadratic, $-\log\sigma((\text{const.}) - \cdot)$. For simplicity, we omit the regularization terms.

| Method | Reference | Loss Function |
|--------|-----------|---------------|
| DPO | Rafailov et al. 2024 | $-\log\sigma(a - b)$ |
| IPO | Azar et al. 2024 | $(a - b - 1)^2$ |
| SPPO | Wu et al. 2024 | $(a - 1/2)^2 + (-b - 1/2)^2$ |
| KTO | Ethayarajh et al. 2024 | $\sigma(-a) + \sigma(b)$ |
| BCO | Jung et al. 2024 | $-\log\sigma(a - \delta) - \log\sigma(-b - \delta)$ |
| DDRO | Ours | $-a - \tilde{b} + (\text{const.})$ |

ing how the preferred data distribution and the unpreferred data distribution interact in the loss.

### 4.1. Estimation error bound

First, we give an estimation error bound of our method with respect to the $L^2$-norm that includes the model misspecification error and a generalization error. Here, let $\Theta$ be the set of parameters of our model and its corresponding set of distributions be $\mathcal{H} = \{p_\theta \mid \theta \in \Theta\}$. Let the *Rademacher complexity* of the model $\mathcal{H}$ be denoted by $\mathcal{R}_n(\mathcal{H})$:

$$\mathcal{R}_n(\mathcal{H}) := \max_{p \in p^+, p^-} \mathbb{E}_{\mathcal{D} \sim p^n}\left[\mathbb{E}_\sigma\left[\sup_{\theta \in \Theta}\left|\frac{1}{n}\sum_{z_i \in \mathcal{D}}\sigma_i p_\theta(z^i)\right|\right]\right],$$

where $(\sigma_i)_{i=1}^n$ is an i.i.d. Rademacher sequence, that is, $P(\sigma_i = 1) = P(\sigma_i = -1) = 1/2$ (see Mohri et al. (2012) for a reference). It is known that the Rademacher complexity measures the variance of the empirical risk minimizer that reflects the complexity of the model. Let $\tilde{p} = \frac{1}{1-t}p_{\text{ref}} - \frac{t}{1-t}p$ for $p \in \mathcal{H}$, and let $h^\pm : \mathbb{R} \to \mathbb{R}$ be $h^+(p) = -f(\tilde{p}/p) + (\tilde{p}/p)f'(\tilde{p}/p)$ and $h^-(p) = -f'(\tilde{p}/p)$. Then, we have the following theorem.

**Theorem 4.1.** *Let $\hat{\theta} = \arg\min_{\theta \in \Theta}\mathcal{L}_{\text{DDRO}}(\theta)$ be the optimal solution of the DDRO loss function with $\gamma = 0$. We assume that there exists some open bounded interval $I \subset \mathbb{R}$ such that $\tilde{p}(y|x)/p(y|x) \in I$ for all $p \in \mathcal{H}$, $x \in \mathcal{X}$, and $y \in \mathcal{Y}$. We also assume that $h^+$ and $h^-$ are Lipschitz continuous on the interval $I$ with a Lipschitz constant $\text{Lip}(h)$. Then, the estimation error of $p_{\hat{\theta}}$ with respect to the true preferred distribution $p^+$ satisfies:*

$$\mathbb{E}_{\mathcal{D}_\pm}\left[\left\|p_{\hat{\theta}} - p^+\right\|_{L^2(p^+)}^2\right] \leq \frac{2(1 - t)^2}{t^2 m_+^2 \mu} \times$$

$$\left[4\text{Lip}(h)(\mathcal{R}_{n^+}(\mathcal{H}) + \mathcal{R}_{n^-}(\mathcal{H})) + \inf_{\theta \in \Theta}\mathcal{L}_{\text{Breg}}(g_\theta)\right],$$

*where the expectation is taken over the realization of the supervised data $\mathcal{D}_\pm$, $m_+ := \min_{x \in \mathcal{X}, y \in \text{supp}(p^+(\cdot|x))} p^+(y \mid x) > 0$, and $\mu := \inf_{s \in I} f''(s) > 0$.*

The proof is given in Appendix C. Since usual Rademacher complexity bounds yield $\mathcal{R}_n(\mathcal{H}) = \mathcal{O}(1/\sqrt{n})$ (Mohri

et al., 2012), the right hand side can be evaluated as $\mathcal{O}\left(\frac{1}{\sqrt{\min\{n^+, n^-\}}} + \inf_{\theta \in \Theta}\mathcal{L}_{\text{Breg}}(g_\theta)\right)$. The term $\inf_{\theta \in \Theta}\mathcal{L}_{\text{Breg}}(g_\theta)$ represents how far the model is misspecified from the true distribution. If the true preferred data distribution is included in the model, this term vanishes. Hence, we see that the bound represents the bias and variance trade-off to estimate an appropriately aligned model. As the data size increases, DDRO becomes more accurate even though we are not putting any assumption on the human preference model.

### 4.2. Connection to existing methods

To elucidate the connection between our DDRO framework and existing methods, we first consider a simplified version of our loss by setting $t = 1/2$ and $\gamma = 0$. Under these conditions, the DDRO loss (4) is reduces to

$$\mathcal{L}_{\text{DDRO}}(\theta) = \mathbb{E}_{p^+}\left[\log 2 - \log\frac{p_\theta}{p_{\text{ref}}}\right] + \mathbb{E}_{p^-}\left[\log 2 - \log\frac{\tilde{p}_\theta}{p_{\text{ref}}}\right].$$

To establish a unified comparison, we follow the single-pair setup from Wu et al. (2024) where only one preference pair $(x, y^+, y^-)$ is available. Define the log-density-ratio terms on the data points $y^+$ and $y^-$ as

$$a = \log\frac{p_\theta(y^+ \mid x)}{p_{\text{ref}}(y^+ \mid x)}, \quad b = \log\frac{p_\theta(y^- \mid x)}{p_{\text{ref}}(y^- \mid x)}.$$

Then, the existing approaches can be summarized in Table 1. We see that existing methods can be expressed as $S(-a + b)$ or $S(-a) + S(b)$ using specific (partially) increasing and (partially) convex functions $S(\cdot)$.

On the other hand, the simplified DDRO can be expressed as $-a - \tilde{b} + (\text{const.})$, where $\tilde{b}$ is defined by

$$\tilde{b} = \log\frac{\tilde{p}_\theta(y^- \mid x)}{p_{\text{ref}}(y^- \mid x)}.$$

The biggest difference from existing methods is that, while existing ones try to *reduce* fitting of $p_\theta$ to unpreferred data by decreasing $b$, DDRO tries to *increase* fitting of $\tilde{p}_\theta$ to the unpreferred data by increasing $\tilde{b}$. Without any assumption

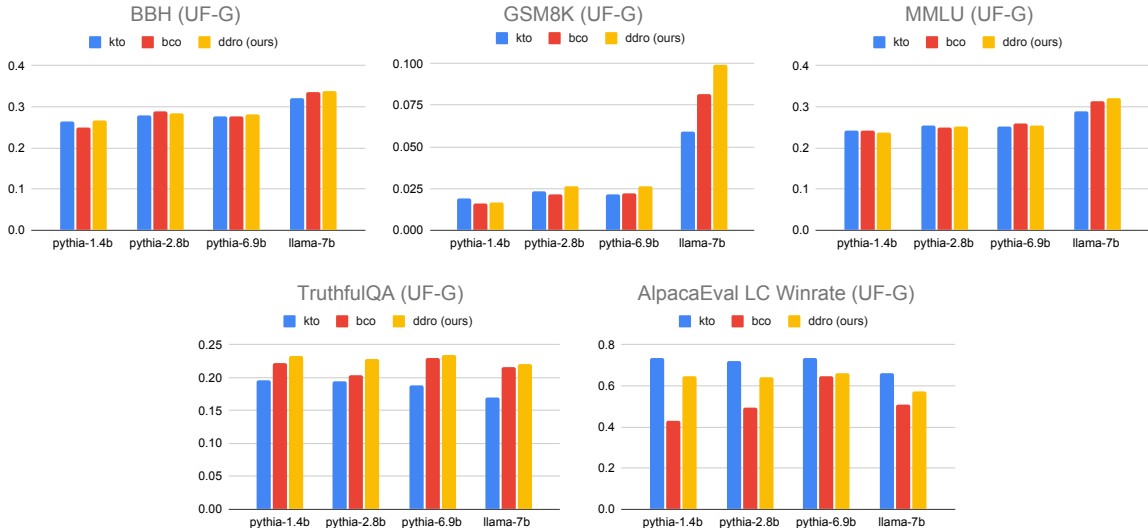

*Figure 1.* Performance comparison of three methods (KTO, BCO, DDRO) on various benchmarks (BBH, GSM8K, MMLU, TruthfulQA, and AlpacaEval LC Winrate). The methods are applied to four different model sizes: Pythia 1.4B, Pythia 2.8B, Pythia 6.9B, and LLaMA 7B, each using the UF-G dataset.

on the human preference model, decreasing the value of $b$ does not necessarily lead to good fit on the preferred data. On the other hand, our method gives good fit of both $p_\theta$ and $\tilde{p}_\theta$ to preferred and unpreferred data respectively, leading to its better explanation of the whole data. This property stems from the proper specification of the loss derived from the Bregman divergence that yields statistical consistency as shown in the previous section.

## 5. Experiments

In this section, we evaluate our proposed method on two datasets: UF-G (unpaired) and UF-B (paired). We first demonstrate that DDRO can achieve performance comparable to or better than existing unpaired methods (KTO, BCO) on UF-G. Next, we investigate how well DDRO can approach the performance of a paired method (DPO) by converting the paired UF-B dataset into an unpaired format for DDRO training. Surprisingly, DDRO remains comparable or superior to DPO on all benchmarks, despite losing information when transforming paired data into unpaired data.

### 5.1. Experimental Setup

**Datasets.** We conduct experiments on two publicly available datasets for LLM alignment: ultrafeedback-gpt-3.5-turbo-helpfulness (UF-G) and ultrafeedback_binarized (UF-B). Both of these datasets are available through `datasets` library (Lhoest et al., 2021). The UF-G dataset is a filtered version of the UltraFeedback dataset (Cui et al., 2023), specifically filtered by helpfulness. After filtering, the UF-G

dataset is transformed into an unpaired dataset. The UF-B dataset is a preference dataset built from the GPT-4 scored UltraFeedback dataset (64k prompts, each with 4 completions). It is created by binarizing the dataset, where the highest scoring completion is labeled as "chosen" and a random completion is labeled as "rejected." This makes UF-B a paired dataset.

**Models.** We train the following base pre-trained LLMs: Pythia (1.4B, 2.8B, 6.9B) (Biderman et al., 2023), LLaMA 7B (Touvron et al., 2023).

**Benchmarks.** We evaluate performance on several downstream benchmarks that assess helpfulness, correctness, or factual consistency. These benchmarks include BBH (Suzgun et al., 2022), which consists of a subset of BIG-Bench (Srivastava et al., 2022) tasks; GSM8K (Cobbe et al., 2021), which features grade-school math problems; MMLU (Hendrycks et al., 2021), a multi-task language understanding benchmark; TruthfulQA (Lin et al., 2022), which evaluates the truthfulness of responses; and AlpacaEval LC Winrate (Dubois et al., 2024), which focuses on length-controlled preference evaluation. We employ `Llama3-70B-Instruct` (AI@Meta, 2024) as an evaluator to determine preferences between responses.

### 5.2. Results on UF-G (Unpaired Dataset)

We first compare DDRO with existing unpaired methods. Figure 1 shows that DDRO achieves performance on par with or exceeding KTO and BCO across four out of five benchmarks. One exception is observed in the AlpacaEval

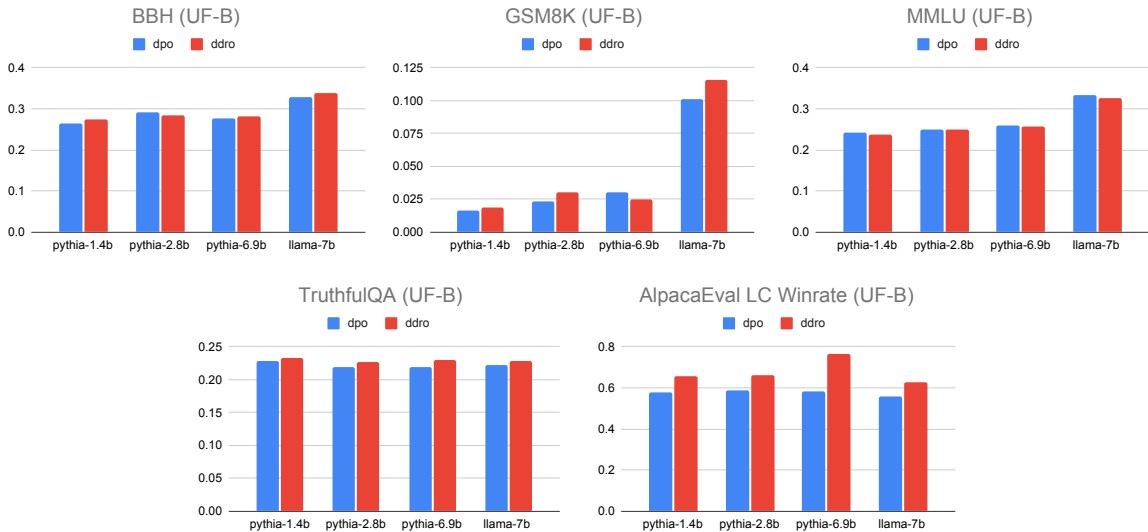

*Figure 2.* Performance comparison of DDRO and DPO on various benchmarks (BBH, GSM8K, MMLU, TruthfulQA, and AlpacaEval LC Winrate). The methods are applied to four different model sizes: Pythia 1.4B, Pythia 2.8B, Pythia 6.9B, and LLaMA 7B, each using the UF-B dataset.

LC Winrate benchmark, where KTO slightly outperforms DDRO. These results confirm that DDRO is highly competitive in the unpaired setting.

### 5.3. Results on UF-B (Paired Dataset) vs. DPO

Next, we examine how effectively DDRO, an unpaired method, can approach or match the performance of a paired method (DPO). For DDRO, we convert the paired samples into unpaired form by taking each prompt-completion pair separately (and discarding explicit pairwise preference links). This artificially creates an unpaired dataset from originally paired data.

Figure 2 presents an interesting result: despite converting a paired dataset into an unpaired form and thus discarding direct pairwise preference information, DDRO consistently performs comparably to or slightly better than DPO across all evaluated benchmarks (BBH, GSM8K, MMLU, TruthfulQA, AlpacaEval). This outcome is noteworthy because DDRO, by design an unpaired approach, manages to achieve performance on par with or exceeding a paired method even under conditions of inherent information loss, suggesting that the direct density ratio estimation technique effectively leverages the remaining preference signals.

### 6. Conclusion

In this paper, we addressed the critical issue of statistical inconsistency in the alignment of large language models (LLMs) and proposed Direct Density Ratio Optimization (DDRO), a novel framework for alignment based on direct

density ratio estimation from data. This framework overcomes the limitations of existing methods that rely on specific preference model assumptions. By directly optimizing the density ratio between preferred and unpreferred distributions using Bregman divergence loss, DDRO theoretically guarantees statistical consistency (Theorem 4.1).

Experimental results on multiple benchmarks (BBH, GSM8K, MMLU, TruthfulQA, and AlpacaEval LC Winrate) confirm that DDRO compares favorably with prior methods, achieving on-par or better performance. It is noteworthy that in comparisons using paired datasets, it achieved results comparable to or better than DPO, despite DPO's ability to leverage preference information.

Furthermore, a key feature of our proposed framework is the flexibility to select the convex function $f$ used in the Bregman divergence. By changing the choice of $f$, we can vary the estimated divergence (Sugiyama et al., 2012b). A more comprehensive investigation of its impact on performance could potentially reveal new benefits. Indeed, (Huang et al., 2024) has shown that preference optimization using the $\chi^2$ divergence as a regularization term instead of the conventional KL divergence offers specific advantages, such as improved robustness.

Overall, our findings highlight the potential of DDRO to address fundamental challenges in LLM alignment. By discarding parametric assumptions on human preferences, DDRO enables truly data-driven alignment, leading to more reliable and human-aligned LLMs.

## Acknowledgements

RH was partially supported by JST CREST (JPMJCR2015). TS was partially supported by JSPS KAKENHI (24K02905) and JST CREST (JPMJCR2115). This research is supported by the National Research Foundation, Singapore, Infocomm Media Development Authority under its Trust Tech Funding Initiative, and the Ministry of Digital Development and Information under the AI Visiting Professorship Programme (award number AIVP-2024-004). Any opinions, findings and conclusions or recommendations expressed in this material are those of the author(s) and do not reflect the views of National Research Foundation, Singapore, Infocomm Media Development Authority, and the Ministry of Digital Development and Information.

## Impact Statement

This paper presents work whose goal is to advance the field of Machine Learning. There are many potential societal consequences of our work, none which we feel must be specifically highlighted here.

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

## A. Proof of Proposition 2.1

*Proof.* We aim to demonstrate the existence of a class of preferences for which no member can be obtained through optimization under the Bradley-Terry model assumption. To this end, let us consider a set of prompts $\mathcal{X} = \{x\}$ and a set of responses $\mathcal{Y} = \{y_a, y_b, y_c\}$.

We define a class of preferences $\mathcal{C}_{\text{pref}}$ parametrized by $t \in [0, 1/2) \cup (1/2, 1]$. Each preference $P_t \in \mathcal{C}_{\text{pref}}$ is characterized by the specific pairwise comparison probabilities:

$$\Pr[y_a \succ y_b|x] = \Pr[y_b \succ y_c|x] = \Pr[y_c \succ y_a|x] = t.$$

If a preference $P_t$ were obtainable, there must exist a reward $r_\phi$ such that

$$\sigma\left(r_\phi(x, y_a) - r_\phi(x, y_b)\right) = \sigma\left(r_\phi(x, y_b) - r_\phi(x, y_c)\right) = \sigma\left(r_\phi(x, y_c) - r_\phi(x, y_a)\right) = t.$$

Here $\sigma$ is a sigmoid function. Due to the strict monotonicity of $\sigma$, it follows that

$$r_\phi(x, y_a) - r_\phi(x, y_b) = r_\phi(x, y_b) - r_\phi(x, y_c) = r_\phi(x, y_c) - r_\phi(x, y_a).$$

Since these differences add up to 0 by canceling each other out, we have

$$r_\phi(x, y_a) - r_\phi(x, y_b) = r_\phi(x, y_b) - r_\phi(x, y_c) = r_\phi(x, y_c) - r_\phi(x, y_a) = 0,$$

which implies $r_\phi(x, y_a) = r_\phi(x, y_b) = r_\phi(x, y_c)$. Then, the model generates a probability:

$$\Pr[y_a \succ y_b|x] = \sigma\left(r_\phi(x, y_a) - r_\phi(x, y_b)\right) = \sigma(0) = 1/2.$$

This contradicts $t \in [0, 1/2) \cup (1/2, 1]$, which indicates that $P_t$ cannot be obtained by optimizing under the Bradley-Terry model assumption. This completes the proof. $\square$

## B. Proof of Proposition 3.3

*Proof.* We are given that $\hat{\theta}$ minimizes the Bregman divergence loss, $\mathcal{L}_{\text{Breg}}(\theta)$, such that $\mathcal{L}_{\text{Breg}}(\hat{\theta}) = 0$. The Bregman divergence loss is defined as $\mathcal{L}_{\text{Breg}}(\theta) = \mathbb{E}_{x \sim p_x, y|x \sim p^+}[\text{Breg}_f(g_\theta \| g^*)]$, where $f$ is a strictly convex function. A fundamental property of the Bregman divergence is that $\text{Breg}_f(g_\theta \| g^*) \geq 0$, with equality holding if and only if $g_\theta = g^*$. Also, since the expected value of a non-negative random variable is zero if and only if the probability of the random variable taking a non-zero value is zero, $g_{\hat{\theta}}(y|x) = g^*(y|x)$ holds $(p^+ \times p_x)$-almost surely. The paper establishes a direct relationship, stating that $p_\theta = p^+$ if and only if $g_\theta = g^*$. Since we have shown $g_{\hat{\theta}} = g^*$ $(p^+ \times p_x)$-almost surely, it follows that $p_{\hat{\theta}}(y|x) = p^+(y|x)$ $(p^+ \times p_x)$-almost surely. This proves the first assertion of the proposition.

Next, we examine the estimator for the unpreferred distribution, $\tilde{p}_\theta$, defined as $\tilde{p}_\theta = \frac{1}{1-t}p_{\text{ref}} - \frac{t}{1-t}p_\theta$. Substituting $\hat{\theta}$ for $\theta$ and using the established result $p_{\hat{\theta}} = p^+$, we obtain $\tilde{p}_{\hat{\theta}}(y|x) = \frac{1}{1-t}p_{\text{ref}}(y|x) - \frac{t}{1-t}p^+(y|x)$. Assumption 3.2 provides the decomposition of the reference distribution as $p_{\text{ref}}(y|x) = tp^+(y|x) + (1-t)p^-(y|x)$. Inserting this into the expression for $\tilde{p}_{\hat{\theta}}$ yields:

$$\tilde{p}_{\hat{\theta}}(y|x) = \frac{1}{1-t}[tp^+(y|x) + (1-t)p^-(y|x)] - \frac{t}{1-t}p^+(y|x) = p^-(y|x).$$

This equality holds $(p^+ \times p_x)$-almost surely, as it relies on the condition $p_{\hat{\theta}}(y|x) = p^+(y|x)$, which itself holds $(p^+ \times p_x)$-almost surely. This demonstrates the second assertion of the proposition.

Thus, we have shown that if $\mathcal{L}_{\text{Breg}}(\hat{\theta}) = 0$, then both $p_{\hat{\theta}}(y|x) = p^+(y|x)$ and $\tilde{p}_{\hat{\theta}}(y|x) = p^-(y|x)$ hold $((p^+ \times p_x) - a.s.)$. $\square$

## C. Proof of Theorem 4.1

*Proof.* First note that, for any $\theta \in \Theta$, we have that

$$\mathcal{L}_{\text{Breg}}(\hat{\theta}) = \mathcal{L}_{\text{Breg}}(\hat{\theta}) - \mathcal{L}_{\text{Breg}}(\theta) + \mathcal{L}_{\text{Breg}}(\theta) + \hat{\mathcal{L}}_{\text{Breg}}(\hat{\theta}) - \hat{\mathcal{L}}_{\text{Breg}}(\hat{\theta}) + \hat{\mathcal{L}}_{\text{Breg}}(\theta) - \hat{\mathcal{L}}_{\text{Breg}}(\theta)$$

$$= \mathcal{L}_{\text{Breg}}(\theta) + \underbrace{(\mathcal{L}_{\text{Breg}}(\hat{\theta}) - \hat{\mathcal{L}}_{\text{Breg}}(\hat{\theta}))}_{(i)} - \underbrace{(\mathcal{L}_{\text{Breg}}(\theta) - \hat{\mathcal{L}}_{\text{Breg}}(\theta))}_{(ii)} + \underbrace{(\hat{\mathcal{L}}_{\text{Breg}}(\hat{\theta}) - \hat{\mathcal{L}}_{\text{Breg}}(\theta))}_{(iii)}. \tag{7}$$

We see that $(iii) \leq 0$ for any $\theta \in \Theta$ because $\hat{\theta}$ minimizes $\hat{\mathcal{L}}_{\mathrm{Breg}}(\theta)$ in the model. Next, we bound the terms $(i)$ and $(ii)$. Since both terms can be bounded as

$$(i), (ii) \leq \sup_{\theta \in \Theta} \left| \hat{\mathcal{L}}_{\mathrm{Breg}}(\theta) - \mathcal{L}_{\mathrm{Breg}}(\theta) \right|,$$

we just need to bound the right hand side. Define the functions $h^+(p; (x, y))$ and $h^-(p; (x, y))$ as

$$h^+(p; (x, y)) = -f\left( \frac{\tilde{p}(y \mid x)}{p(y \mid x)} \right) + \frac{\tilde{p}(y \mid x)}{p(y \mid x)} f'\left( \frac{\tilde{p}(y \mid x)}{p(y \mid x)} \right),$$
$$h^-(p; (x, y)) = -f'\left( \frac{\tilde{p}(y \mid x)}{p(y \mid x)} \right).$$

Then, we have that

$$
\mathbb{E}_{\mathcal{D}_\pm} \sup_{\theta \in \Theta} \left| \hat{\mathcal{L}}_{\mathrm{Breg}}(\theta) - \mathcal{L}_{\mathrm{Breg}}(\theta) \right| = \mathbb{E}_{\mathcal{D}_\pm} \left[ \sup_{p \in \mathcal{H}} \left| \frac{1}{n^+} \sum_{z^+ \in \mathcal{D}_+} h^+(p; z^+) - \mathbb{E}_{p^+}\left[ h^+(p) \right] \right. \right.
$$
$$
\left. \left. + \frac{1}{n^-} \sum_{z^- \in \mathcal{D}_-} h^-(p; z^-) - \mathbb{E}_{p^-}\left[ h^-(p) \right] \right| \right]
$$
$$
\leq \mathbb{E}_{\mathcal{D}_+} \left[ \sup_{p \in \mathcal{H}} \left| \frac{1}{n^+} \sum_{z_i^+ \in \mathcal{D}_+} h^+(p; z_i^+) - \mathbb{E}_{p^+}\left[ h^+(p) \right] \right| \right]
$$
$$
+ \mathbb{E}_{\mathcal{D}_-} \left[ \sup_{p \in \mathcal{H}} \left| \frac{1}{n^-} \sum_{z_i^- \in \mathcal{D}_-} h^-(p; z_i^-) - \mathbb{E}_{p^-}\left[ h^-(p) \right] \right| \right].
$$

Here, by the standard symmetrization trick (Theorem 3.1 of Mohri et al. (2012)) and the contraction inequality (Theorem 11.6 of Boucheron et al. (2013) or Theorem 4.12 of Ledoux & Talagrand (1991)), we have

$$
\mathbb{E}_{\mathcal{D}_\pm} \left[ \sup_{p \in \mathcal{H}} \left| \frac{1}{n^\pm} \sum_{z_i^\pm \in \mathcal{D}_\pm} h^\pm(p; z_i^\pm) - \mathbb{E}_{p^\pm}\left[ h^\pm(p) \right] \right| \right] \leq 2\mathcal{R}_{n^\pm}(h^\pm \circ \mathcal{H}) \leq 2\mathrm{Lip}(h)\mathcal{R}_{n^\pm}(\mathcal{H}).
$$

Finally, the left hand side of Eq. (7) can be lower bounded by

$$\frac{\|p_{\hat{\theta}} - p^+\|^2_{L^2(p^+)}}{\frac{2(1-t)^2}{t^2 m_+^2 \mu}} \leq \mathcal{L}_{\mathrm{Breg}}(\hat{\theta}),$$

from Lemma C.1. Finally, by taking infimum over $\theta \in \Theta$ in the right hand side of Eq. (7), we obtain the assertion. $\square$

### C.1. Auxiliary lemma

**Lemma C.1.** *Suppose that the same assumption as Theorem 4.1 holds. For any $p \in \mathcal{H}$, there exists $\mu > 0$ such that*

$$\|p - p^+\|^2_{L^2(p^+)} \leq \frac{2(1-t)^2}{t^2 m_+^2 \mu} \mathcal{L}_{\mathrm{Breg}}(g)$$

*where $m_+ := \min_{x \in \mathcal{X}, y \in \mathrm{supp}(p^+(\cdot \mid x))} p^+(y \mid x) > 0$.*

*Proof.* Lemma 4 in Kato & Teshima 2021 states that for any $p \in \mathcal{H}$, there exists $\mu > 0$ such that

$$\|g - g^*\|^2_{L^2(p^+)} \leq \frac{2}{\mu} \mathcal{L}_{\mathrm{Breg}}(g) \tag{8}$$

We now demonstrate that when $g$ and $g^*$ are close, $p$ and $p^+$ are also close. Let $x \in \mathcal{X}$ be an arbitrary prompt. For notational simplicity, we omit the conditioning on $x$.

$$\|g - g^*\|^2_{L^2(p^+)} = \left\| \frac{\tilde{p}_\theta}{p} - \frac{p^-}{p^+} \right\|^2_{L^2(p^+)}$$

$$= \left\| \frac{p_{\text{ref}}}{(1-t)pp^+} \left( p - p^+ \right) \right\|^2_{L^2(p^+)}$$

Due to the finiteness of $\mathcal{Y}$, $p^+$ has a positive minimum value on its support:

$$m_+ := \min_{y \in \mathcal{Y}} p^+(y) > 0$$

Furthermore, by Assumption 3.2, on the support of $p^+$, we have $p_{\text{ref}} \geq tp^+ \geq tm_+$. Consequently:

$$\|g - g^*\|^2_{L^2(p^+)} \geq \left\| \frac{p_{\text{ref}}}{(1-t)pp^+} \left( p - p^+ \right) \right\|^2_{L^2(p^+)}$$

$$\geq \left\| \frac{tm_+}{(1-t)pp^+} \left( p - p^+ \right) \right\|^2_{L^2(p^+)}$$

$$\geq \frac{t^2 m_+^2}{(1-t)^2} \left\| p - p^+ \right\|^2_{L^2(p^+)}$$

By combining this with (8), we obtain that

$$\frac{t^2 m_+^2}{(1-t)^2} \left\| p - p^+ \right\|^2_{L^2(p^+)} \leq \frac{2}{\mu} \mathcal{L}_{\text{Breg}}(g).$$

Rearranging terms, we find that

$$\left\| p - p^+ \right\|^2_{L^2(p^+)} \leq \frac{2(1-t)^2}{t^2 m_+^2 \mu} \mathcal{L}_{\text{Breg}}(g).$$

$\square$

## D. Training Details

This section provides details regarding the training procedures employed for each model in our experiments.

**Training Setup:** All models were trained using the TRL library (von Werra et al., 2020). The batch size was set to 64, and training was conducted for 1 epoch across all methods and datasets. The AdamW optimizer (Loshchilov & Hutter, 2017) was used for optimization.

**Learning Rates:** The learning rates for each method were as follows: 5e-7 for DPO, 5e-6 for KTO, 1e-6 for BCO, and 5e-7 for DDRO. For existing methods, we utilized the learning rate values reported in their respective papers (Rafailov et al., 2024; Ethayarajh et al., 2024; Jung et al., 2024). It is important to note that the DDRO learning rate may not be optimal for every choice of the set $S$. Since the scale of the loss function can vary depending on the selection of $S$, the learning rate might need to be adjusted accordingly.

**Other Details:** Following the approach in Ethayarajh et al. 2024, we did not include the KL regularization term in the gradient calculation for training stability.

## E. Impact of the Choice of Smoothing Function $S(x)$ on Training Stability

To analyze in detail the impact of the choice of the smoothing function $S(x)$, which was introduced in the loss function of our proposed method DDRO (Equation (6)) to encourage stable learning, we conducted an ablation study. Theoretically, $S(x)$ is expected to smooth the Bregman Divergence term in the loss function and stabilize the optimization process. As described in Section 3.3, while $S(x)$ should ideally satisfy monotonicity and convexity from a theoretical perspective, practically, it is not always necessary to adhere strictly to these constraints, allowing for the exploration of various functional forms.

We compared four functional forms for $S(x)$. As a baseline without smoothing, we used the identity function ($S(x) = x$). We also tested $S(x) = \sigma(x), \log \sigma(x), -\log \sigma(-x)$.

Figure 3 illustrates the transition of the gradient norm during training for each choice of $S(x)$. The figure reveals that with the identity function ($S(x) = x$), gradient spikes occur during training, suggesting unstable learning dynamics. In contrast, when using the other functions, the gradient spikes are suppressed, confirming more stable training.

These results demonstrate the crucial role of smoothing with $S(x)$ for the stable learning of DDRO. The gradient spikes observed with the identity function ($S(x) = x$) indicate that without smoothing, the loss landscape may become complex, potentially leading to convergence to suboptimal solutions or unstable learning. Conversely, we observed stabilization of learning with any choice of $S$ other than the identity function.

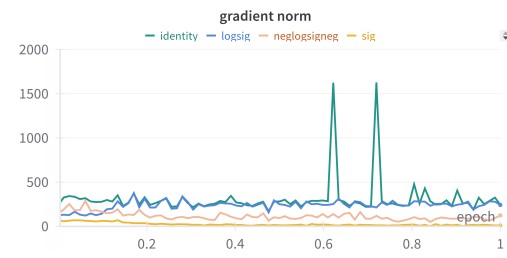

*Figure 3.* Gradient norm during training for different smoothing functions $S(x)$ (identity: $S(x) = x$, logsig: $S(x) = \log \sigma(x)$, neglogsigneg: $S(x) = -\log \sigma(-x)$, sig: $S(x) = \sigma(x)$).

## F. Impact of the Choice of Convex Function $f$ in Bregman divergence on Benchmark Performance

In our proposed framework (5), there remains flexibility in the choice of the convex function $f$ used for the Bregman divergence. This section investigates how performance on various benchmarks changes when $f$ is altered.

For the experimental setup, we trained Pythia 1.4B (Biderman et al., 2023) on the UF-G dataset (see Section 5) with different convex functions $f$. The candidates for $f$, referencing (Sugiyama et al., 2012b), are as follows:

$$f(t) = t \log t - (1 + t) \log(1 + t), \quad t \log t - t, \quad (t-1)^2/2,$$
$$(t^{1+\alpha} - t)/\alpha \quad (\alpha = 0.1,\ 0.5,\ 1.0,\ 2.0,\ 5.0)$$

The learning rate was individually tuned for each function $f$ by selecting from 5e-7, 5e-8, or 5e-9 to accommodate the variation in loss scaling.

The results are presented in Table 2. We note that $f(t) = t \log t - (1 + t) \log(1 + t)$, listed in the first row of the table, is the function used in the main experiments of this paper, and its corresponding results are identical to those shown in Figure 1 and Table 3. As indicated in Table 2, AlpacaEval was the benchmark where the choice of $f$ led to the most remarkable differences in performance. For the other benchmarks, the performance variations were generally within a range of 1-2%. Notably, the function employed in the main experiments of this paper demonstrated superior performance over the other functions in AlpacaEval, which supports the validity of our choice.

*Table 2.* Benchmark performance of Pythia 1.4B with varying convex functions $f(t)$ for Bregman divergence. Scores are reported for BBH, GSM8K, MMLU, TruthfulQA, and AlpacaEval, alongside the learning rate (lr) used for each function. The function $f(t) = t \log t - (1 + t) \log(1 + t)$ achieved the highest AlpacaEval score (bolded), which is adopted in our main experiment.

| $f(t)$ | | lr | BBH | GSM8K | MMLU | TruthfulQA | AlpacaEval |
|---|---|---|---|---|---|---|---|
| $t \log t - (1 + t) \log(1 + t)$ | | 5e-7 | 0.2593 | 0.0167 | 0.2416 | 0.2313 | **0.5836** |
| $t \log t - t$ | | 5e-9 | 0.2594 | 0.0227 | 0.2426 | 0.2264 | 0.4962 |
| $(t - 1)^2/2$ | | 5e-9 | 0.2603 | 0.0235 | 0.2423 | 0.2264 | 0.5000 |
| | $(\alpha = 0.1)$ | 5e-9 | 0.2591 | 0.0227 | 0.2425 | 0.2264 | 0.4977 |
| | $(\alpha = 0.5)$ | 5e-8 | 0.2593 | 0.0220 | 0.2428 | 0.2264 | 0.4872 |
| $(t^{1+\alpha} - t)/\alpha$ | $(\alpha = 1.0)$ | 5e-8 | 0.2599 | 0.0174 | 0.2436 | 0.2301 | 0.4742 |
| | $(\alpha = 2.0)$ | 5e-9 | 0.2589 | 0.0220 | 0.2425 | 0.2264 | 0.4955 |
| | $(\alpha = 5.0)$ | 5e-8 | 0.2597 | 0.0205 | 0.2426 | 0.2264 | 0.4820 |

# G. Benchmark Performance Details

This section provides the detailed numerical results for the experiments presented in Section 5. We report performance metrics on five standard benchmarks: BBH, GSM8K, MMLU, TruthfulQA, and AlpacaEval (LC Winrate). The evaluated methods include DPO, KTO, BCO, and our method DDRO and we applied them to Pythia and LLaMA model families.

In Table 3 and Table 4, for each model and benchmark combination, the highest score among the compared methods is indicated in **bold**.

*Table 3.* Comparison of KTO, BCO, and DDRO performance across benchmarks on training models on UF-G dataset (unpaired). DDRO is comparable to or outperforms other methods on all evaluated benchmarks except for AlpacaEval.

| Models | | BBH | GSM8K | MMLU | TruthfulQA | AlpacaEval |
|---|---|---|---|---|---|---|
| Pythia 1.4B | + KTO | **0.2634** | **0.0190** | **0.2432** | 0.1958 | **0.7364** |
| | + BCO | 0.2503 | 0.0159 | 0.2422 | 0.2228 | 0.4284 |
| | + DDRO | 0.2593 | 0.0167 | 0.2416 | **0.2313** | 0.5836 |
| Pythia 2.8B | + KTO | 0.2797 | 0.0235 | **0.2539** | 0.1946 | **0.7214** |
| | + BCO | 0.2877 | 0.0212 | 0.2504 | 0.2032 | 0.4933 |
| | + DDRO | **0.2854** | **0.0265** | 0.2532 | **0.2240** | 0.5740 |
| Pythia 6.9B | + KTO | 0.2758 | 0.0212 | 0.2526 | 0.1885 | **0.7341** |
| | + BCO | 0.2755 | 0.0220 | 0.2590 | 0.2301 | 0.6469 |
| | + DDRO | **0.2794** | **0.0265** | **0.2531** | **0.2338** | 0.5986 |
| LLaMA 7B | + KTO | 0.3213 | 0.0591 | 0.2877 | 0.1701 | **0.6636** |
| | + BCO | **0.3345** | 0.0819 | 0.3123 | 0.2166 | 0.5108 |
| | + DDRO | 0.3330 | **0.0993** | **0.3182** | **0.2191** | 0.5280 |

*Table 4.* Comparison of DPO and DDRO performance across benchmarks on training models on UF-B dataset (paired). DDRO is comparable to or outperforms DPO regardless of the limitation that it cannot directly leverage the paired structure of the data.

| Models | | BBH | GSM8K | MMLU | TruthfulQA | AlpacaEval |
|---|---|---|---|---|---|---|
| Pythia 1.4B | + DPO | 0.2648 | 0.0167 | **0.2423** | 0.2277 | 0.5785 |
| | + DDRO | **0.2715** | **0.0182** | 0.2396 | **0.2277** | **0.6609** |
| Pythia 2.8B | + DPO | 0.2907 | **0.0235** | 0.2485 | 0.2191 | 0.5886 |
| | + DDRO | **0.2910** | 0.0205 | **0.2582** | **0.2326** | **0.6695** |
| Pythia 6.9B | + DPO | 0.2766 | **0.0303** | **0.2582** | 0.2191 | 0.5804 |
| | + DDRO | **0.2892** | 0.0296 | 0.2542 | **0.2375** | **0.6405** |
| LLaMA 7B | + DPO | 0.3281 | **0.1008** | **0.3332** | 0.2215 | 0.5564 |
| | + DDRO | **0.3350** | 0.1001 | 0.3220 | **0.2240** | **0.5973** |

