# OpenReview forum: "Direct Density Ratio Optimization: A Statistically Consistent Approach to Aligning Large Language Models"
_ICML.cc/2025/Conference — ICML 2025 poster_

### Official Review · Reviewer_KCaR · 2025-03-04

**Overall Recommendation:** 2

**Summary:**

This paper introduces a new offline reinforcement learning approach for LLM alignment, motivated by the lack of statistical consistency in the traditional BT model, which underpins popular offline algorithms such as DPO. The author proposes directly optimizing the density ratio using a Bregman divergence loss and proves that the estimated parameters exhibit good statistical consistency under certain assumptions. Experiments conducted on Ultrafeedback datasets show marginal improvements over existing offline RL methods.

**Claims And Evidence:**

The primary claim is well supported by the theoretical analysis.

**Essential References Not Discussed:**

Most of the related papers are well cited.

**Experimental Designs Or Analyses:**

The experiments are well-designed, but the performance of method indicates that the work is incremental to the research of alignment.

**Methods And Evaluation Criteria:**

The proposed method is theoretically sound, and the empirical studies are well-designed. However, my concern lies in its performance. While the work establishes a solid theoretical foundation for offline RL methods, the proposed approach does not demonstrate compelling advantages over existing models, which significantly limits its practical applicability.

**Other Comments Or Suggestions:**

The draft needs further proof-reading.
For example:
Line 165: Here we consider a situation where he data...-> `he' must be `the'
Line 358: the citation for MMLU is a '?'

**Other Strengths And Weaknesses:**

see my comments above

**Questions For Authors:**

N/A

**Relation To Broader Scientific Literature:**

The proposed model could be a valuable addition to the family of offline RL methods. The authors clearly establish connections with existing approaches and provide solid theoretical analysis, which is often lacking in prior work。

**Theoretical Claims:**

Yes, I checked the proof shown in Appendix.

---

> ### Author Rebuttal · Authors · 2025-04-01
>
> Thank you for your review and feedback on our paper. We address your points below.
>
> 1.  **Performance concerns:**
>     The reviewer expressed concern that DDRO does not demonstrate compelling advantages over existing models, potentially limiting its practical applicability. Regarding the magnitude of improvement, it's important to evaluate this within the context of single-iteration preference optimization. We believe that achieving dramatic performance gains from just one optimization step is challenging. In fact, this tendency for incremental progress is common in the field (see works such as ORPO[1], CPO[2], and SPPO[3], which often report similar levels of improvement after one optimization step).
>
>     Considering this background – where substantial leaps are not always expected from a single iteration – we believe DDRO's results are noteworthy. We highlight three key points demonstrating its effectiveness:
>
>     - Broad Consistency: DDRO achieves the best performance on the majority of benchmarks and model sizes tested. This result could indicate that explicitly modeling preferences causes a failure to model the preferences necessary for many benchmarks, limiting good performance only to those benchmarks where the model luckily matches the required preferences.
>
>     - Significant Gains on Specific Tasks: On specific benchmarks like GSM8K with Llama-7b, DDRO shows substantial improvements over KTO (over 50\% relative gain) and BCO (over 20\% relative gain), which we believe are non-trivial, especially in the context where large gaps are not always expected after one iteration of alignment.
>
>     - Effectiveness in Paired Settings: DDRO achieves performance better than DPO in the paired setting, despite discarding the comparison information that DPO utilizes. This underscores the effectiveness of DDRO.
>
>     References:
>
>     - [1] Hong, J., Lee, N., & Thorne, J. (2024). ORPO: Monolithic preference optimization without reference model. arXiv preprint arXiv:2403.07691.
>     - [2] Xu, H., Sharaf, A., Chen, Y., Tan, W., Shen, L., Van Durme, B., ... & Kim, Y. J. (2024). Contrastive preference optimization: Pushing the boundaries of llm performance in machine translation. arXiv preprint arXiv:2401.08417.
>     - [3] Wu, Y., Sun, Z., Yuan, H., Ji, K., Yang, Y., & Gu, Q. (2024). Self-play preference optimization for language model alignment. arXiv preprint arXiv:2405.00675.
>
> 2.  **Proofreading errors:**
>     Thank you for pointing out the specific typo ("he" -> "the" in Line 165) and the broken MMLU citation (Line 358). We apologize for these oversights. We will meticulously proofread the manuscript, correcting these and any other errors, before submitting the camera-ready version.
>
>
> We appreciate your feedback and hope these clarifications address your concerns about DDRO's performance and practical relevance.

---

> > ### Comment · Reviewer_KCaR · 2025-04-02
> >
> > Thanks for the reply. I would like to maintain my rating. One additional suggestion: it is better to present evaluation results with tables. It is very difficult to recognize the performance (and thus recognize the practical metrits of the proposed method) through the bar charts.

---

> > > ### Author Response · Authors · 2025-04-04
> > >
> > > Thank you for your reply and the additional constructive suggestion regarding the presentation of our results.
> > >
> > > We completely agree that presenting the evaluation results in tables will improve clarity and make it easier to assess the performance differences and practical merits of our proposed method compared to the bar charts. We appreciate this feedback and will incorporate tables in the revised version as suggested.
> > >
> > > We believe this clearer tabular format will help demonstrate the significance of the improvements achieved by DDRO.
> > > To illustrate this, here is an excerpt of how the results will be presented.
> > >
> > > | Base Model  | TruthfulQA KTO | TruthfulQA BCO | TruthfulQA DDRO | GSM8K KTO | GSM8K BCO | GSM8K DDRO |
> > > |----------------|----------------|----------------|----------------|----------------|----------------|----------------|
> > > | Pythia 1.4B | 0.1958         | 0.2228         | **0.2326** | **0.0190** | 0.0159    | 0.0167            |
> > > | Pythia 2.8B | 0.1946         | 0.2032         | **0.2277** | 0.0235    | 0.0212    | **0.0265** |
> > > | Pythia 6.9B | 0.1885         | 0.2301         | **0.2338** | 0.0212    | 0.0220    | **0.0265** |
> > > | LLaMA 7B    | 0.1701         | 0.2166         | **0.2203** | 0.0591    | 0.0819    | **0.0993** |
> > >
> > > As the table demonstrates, DDRO achieves consistent improvements over the compared methods.
> > > Based on calculations from the table, it shows that DDRO achieves a 1.050x improvement (average of 1.044x, 1.120x, 1.016x, 1.017x) over the second-best on TruthfulQA and a 1.107x improvement (average of 0.88x, 1.129x, 1.207x, 1.121x) over the second-best performing method on GSM8K. We believe these average gains are significant, especially considering that the improvement reached as high as 1.207x in the best-performing case compared to the second-best method.
> > >
> > > Furthermore, we understand your concern regarding practical applicability stemming from the perceived magnitude of improvement. We would like to place these results in the context of the highly competitive and challenging field of single-iteration preference optimization. Incremental gains are often the standard, even in state-of-the-art research recognized by the community:
> > >
> > > 1.  **DiscoPOP (NeurIPS 2024):** When training the zephyr-7b-gemma-sft model, DiscoPOP achieved an LC winrate of 65.18, representing a 1.029x improvement compared to training the same base model with DPO (which achieved 63.34).
> > > 2.  **Iterative RPO (NeurIPS 2024):** Similarly, training Llama-2-70b-chat with Iterative RPO resulted in an ARC-Challenge score of 84.8, a 1.024x improvement over training the same base model with DPO (which scored 82.8).
> > > 3.  **XPO (implemented in the popular trl library):** Comparing the first iteration of training the Llama-3-8B-Flow-SFT model, XPO-iter1 (score: 63.1) showed a 1.0014x gain on MMLU compared to DPO-iter1 (score: 63.01) applied to the same base model. On other benchmarks for the same comparison, XPO-iter1 performed slightly worse than DPO-iter1 (GSM8K: 75.97 vs 77.79, 0.977x; LC winrate: 22.14 vs 23.27, 0.952x).
> > >
> > > These examples demonstrate that the improvements achieved by DDRO are not only comparable to, but in some cases exceed, the gains reported in highly-regarded works accepted at top venues (NeurIPS 2024) or integrated into widely-used libraries (trl). This strongly suggests that such improvements are considered significant contributions within the research community and are indicative of practical value, especially given the inherent difficulty of achieving large leaps in performance with single-iteration alignment techniques.
> > >
> > > We hope that this clearer tabular presentation, combined with the provided context comparing DDRO's gains to recognized SOTA methods, will facilitate a positive reassessment of the practical merits and significance of our work.
> > >
> > > Thank you once again for your thorough review and valuable feedback, which are helping us strengthen the paper.
> > >
> > > References:
> > > 1.  Lu, C., Holt, S., Fanconi, C., Chan, A., Foerster, J., van der Schaar, M., & Lange, R. (2024). Discovering preference optimization algorithms with and for large language models. *Advances in Neural Information Processing Systems, 37*, 86528-86573.
> > > 2.  Pang, R. Y., Yuan, W., He, H., Cho, K., Sukhbaatar, S., & Weston, J. (2024). Iterative reasoning preference optimization. *Advances in Neural Information Processing Systems, 37*, 116617-116637.
> > > 3.  Xie, T., Foster, D. J., Krishnamurthy, A., Rosset, C., Awadallah, A., & Rakhlin, A. (2024). Exploratory preference optimization: Harnessing implicit q*-approximation for sample-efficient rlhf. *arXiv preprint arXiv:2405.21046*.

---

### Official Review · Reviewer_VDcC · 2025-03-14

**Overall Recommendation:** 2

**Summary:**

This paper introduces Direct Density Ratio Optimization (DDRO), a novel alignment method for large language models (LLMs) that addresses the statistical inconsistency of existing approaches reliant on restrictive preference models (e.g., Bradley-Terry). By directly estimating the density ratio between preferred and unpreferred output distributions, DDRO bypasses explicit preference modeling, ensuring statistical consistency—guaranteed convergence to true human preferences as data scales, regardless of the underlying preference structure. Theoretical analysis validates this property, while experiments across some benchmarks demonstrate the effectiveness of DDRO.

**Claims And Evidence:**

Yes

**Essential References Not Discussed:**

No.

**Experimental Designs Or Analyses:**

Yes

**Methods And Evaluation Criteria:**

Yes

**Other Comments Or Suggestions:**

Please refer to Strengths and Weakness part.

**Other Strengths And Weaknesses:**

### Strength

1. The paper is well written and easy to follow.

2. This paper studies a necessary problem of direct alignment algorithms on unpaired data.

3. The proposed DDRO is reasonable and provide a novel perspective.

### Weakness

1. The author compares their empirical results of DDRO with KTO. The improvement seems limited. Can authors provide average results on all datasets to better demonstrate the effectiveness of DDRO.

2. [1] also discusses the density ratio estimation. Can authors discuss the main difference and advantage of their method?

3. In the experiment section, it's better for the author to compare with the state-of-the-art method like SPPO to further verify the effectiness of DDRO.

[1] How to Leverage Demonstration Data in Alignment for Large Language Model? A Self-Imitation Learning Perspective.

**Questions For Authors:**

Please refer to Strengths and Weakness part.

**Relation To Broader Scientific Literature:**

This paper studies the problem of direct alignment algorithms for unpaired data which is practical and necessary.

**Theoretical Claims:**

Yes

---

> ### Author Rebuttal · Authors · 2025-04-01
>
> Thank you for reviewing our paper and providing valuable feedback. We address your points below.
>
>
> 1.  **Limited improvement and average results:**
>     The reviewer observed that the empirical improvement of DDRO over KTO seems limited and suggested providing average results across datasets. Regarding the magnitude of improvement, we would like to place this in the context of one-iteration preference optimization. It is generally understood that achieving dramatically large performance gains from a single iteration of alignment can be challenging. This characteristic of incremental progress is commonly observed in the field, as evidenced by recent works like ORPO, CPO, and SPPO, which also often report performance improvements of a similar nature after one optimization step. Viewed within this context – where substantial leaps are not always expected from one iteration – we believe DDRO demonstrates notable effectiveness. We wish to highlight three key points:
>
>     Consistency: DDRO achieves the best performance on the majority of benchmarks and model sizes tested.
>
>     Significant Gains: On specific benchmarks like GSM8K with Llama-7b, DDRO shows substantial improvements over KTO (over 50\% relative gain) and BCO (over 20\% relative gain), which we believe are non-trivial, especially in the context where large gaps are not always expected after one iteration of alignment.
>
>     Paired Setting: DDRO achieves performance better than DPO in the paired setting, despite discarding the comparison information that DPO utilizes. This underscores the effectiveness of DDRO.
>
>     Regarding averaging results: While technically feasible, averaging scores across different metrics and scales can be misleading. Metrics with smaller variance or lower chance rates might be unjustly de-emphasized. We believe presenting the full suite of results, as in Figure 1, provides a more transparent and informative picture of DDRO's broad effectiveness, rather than potentially obscuring performance variations through averaging.
>
>
>
> 2.  **Comparison with GSIL [1]:**
>     Thank you for your comment regarding GSIL. While both GSIL and our method use density ratios, their objectives and approaches are fundamentally different.
>     First, construction of the objective is different. GSIL aims to minimize $\mathrm{KL}(\pi_{\text{data}}) \| (\pi_\theta)$, effectively making the density ratio $(\pi_{data} / \pi_\theta)$ approach 1. Our method, however, aims to match the policy density ratio $(p_\theta / p_{ref})$ to the true density ratio $(p_+ / p_{ref})$. Thus, our core objectives differ significantly.
>     Second, GSIL requires multiple iterations to estimate $\pi_{\text{data}}$, and its theoretical convergence is not guaranteed. In contrast, our method runs a single iteration, offering advantages in efficiency and theoretical tractability.
>     We hope this clarifies the distinction.
>
>
>
>
>
> 3.  **Comparison with state-of-the-art methods like SPPO:**
>     We appreciate the suggestion to compare against state-of-the-art methods. However, our work primarily focuses on the setting where offline, binary feedback labels are available (i.e., unpaired preference data). Methods like SPPO operate under a different, richer setting. SPPO typically requires access to a pre-trained reward model, allowing it to use continuous reward scores, and often involves online interaction or scoring during training. Comparing DDRO directly with methods designed for such richer information settings might not be entirely appropriate, as the available data and feedback mechanisms are fundamentally different. Our goal was to develop a theoretically sound and practically effective method specifically for the common scenario of offline binary feedback, where methods like SPPO might not be directly applicable or would require significant adaptation.
>
>
> Thank you again for your comments, which encourage us to clarify the positioning and strengths of DDRO.

---

### Official Review · Reviewer_v1jw · 2025-03-14

**Overall Recommendation:** 4

**Summary:**

This paper introduces Direct Density Ratio Optimization (DDRO), a method for aligning LLMs by directly estimating the density ratio between preferred and unpreferred output distributions. DDRO minimizes a Bregman divergence-based loss, eliminating the need for explicit preference modeling while ensuring statistical consistency. Empirical results show that DDRO performs on par or outperforms existing alignment methods in two settings: (1) unpaired preference data, compared to KTO and BCO, and (2) paired preference data, compared to DPO.

**Claims And Evidence:**

The claims made in Section 1 of the paper are well supported by the methodology design, theoretical analysis, and empirical results.

**Essential References Not Discussed:**

The paper discusses and compares most key methods in LLM alignment, as summarized in Table 1. One widely adopted preference optimization method, ORPO, is not cited. While ORPO is a reference-model-free approach and may not be directly comparable to DDRO, discussing its relevance could potentially provide additional context.

Hong, J., Lee, N., & Thorne, J. (2024). Orpo: Monolithic preference optimization without reference model. arXiv preprint arXiv:2403.07691.

**Experimental Designs Or Analyses:**

**Strengths**

1. The evaluation across five benchmarks covering different language tasks is comprehensive.
2. The experiment results are generally strong and support the claim that DDRO performs on par with or outperforms baseline alignment methods on both unpaired and paired preference data.

**Questions**

1. In Section 5.2, Figure 1 suggests that DDRO outperforms KTO on GSM8K for most model sizes, which appears inconsistent with the description in Section 5.2. Additionally, DDRO underperforms KTO across all model sizes on AlpacaEval. Could you provide insights into why DDRO might specifically underperform on AlpacaEval?

**Methods And Evaluation Criteria:**

**Strengths**

1. Direct density ratio estimation is well-motivated for LLM alignment, as it eliminates the need for an explicit preference model which may not accurately reflect human preferences. It also eliminates the reliance on paired preference data, which can be costly to annotate.

2. The formulation of DDRO is well-structured, and its derivation is clear and easy to follow.

3. DDRO models are trained using UltraFeedback, a widely used public preference dataset for LLM alignment. The aligned LLMs are evaluated on five diverse benchmarks covering language understanding and generation tasks, making the evaluation setup comprehensive.

**Questions**

1. In more realistic settings, when human labelers annotate single generations without direct comparison, many responses may be classified as neutral—neither strongly preferred nor unpreferred. Could DDRO handle such neutral data?

2. DDRO assumes a constant $p(+ | x)$. What would be a good way to determine $p(+ | x)$? How robust is DDRO to policies with different $p(+ | x)$?

**Other Comments Or Suggestions:**

1. In Equation 6, the KL term for the preferred examples appears to be mistakenly written as the KL term for the unpreferred ones.
2. In Section 5.1, the reference to the MMLU benchmark is broken.

**Other Strengths And Weaknesses:**

Please refer to previous comments.

**Questions For Authors:**

Please refer to previous comments.

**Relation To Broader Scientific Literature:**

LLM alignment is an important research direction for ensuring the broad utility and safe deployment of LLMs. This paper contributes to this field by introducing a preference-modeling-free approach, which avoids making parametric assumptions about human preferences. By directly inferring the preference distribution from data, the proposed method alleviates the need for explicitly paired preference data, reducing annotation costs and improving scalability. This provides a more generalizable and flexible framework for LLM alignment for broader applications.

**Theoretical Claims:**

I briefly checked the proof of Theorem 4.1 in Appendix A without verifying all details. The theorem’s insights on the bias-variance trade-off are interesting.

I have a question regarding the bias term—could you provide some intuitive examples of scenarios where the model $g_\theta$ might be misspecified? Given that data are generated from the reference model and subsequently annotated by humans, and that $g_\theta$ is initialized from the same reference policy, what are the cases where $g_\theta$ might fail to capture the true preference distribution? Additionally, are there ways to mitigate such misspecification through strategies like prompt selection?

---

> ### Author Rebuttal · Authors · 2025-04-01
>
> Thank you for your insightful comments and questions regarding our work on DDRO. We appreciate the careful reading and valuable feedback.
>
>
>
> 1.  **Handling neutral data:**
>     The reviewer asked if DDRO can handle "neutral" labels, which might arise when annotators evaluate single responses without direct comparison. Currently, DDRO is formulated to utilize binary preference signals (preferred/unpreferred) and it cannot directly incorporate a distinct "neutral" category. However, we believe that binary feedback collection is a prevalent and practical scenario. Major LLM services like Gemini, Claude, and ChatGPT often employ simple thumbs-up/thumbs-down mechanisms to collect data, which correspond directly to the binary setting DDRO addresses. Therefore, we consider the setting DDRO operates in to be sufficiently realistic and widely applicable.
>
>
>
> 2.  **Determining $p(+|x)$ and robustness:**
>     The reviewer inquired about determining the constant $p(+|x)$ and DDRO's robustness to its value. We assumed $p(+|x)$ to be constant (denoted as $t$) for simplicity in our analysis. This constant $t$ can be empirically estimated by sampling responses from the reference policy $p_{\text{ref}}$ for a set of prompts $x$ and calculating the proportion of responses deemed "preferred". However, our empirical investigations (which we can add details of in the final version) showed that the training dynamics and final performance of DDRO were not highly sensitive to the specific value chosen for $t$. Using a standard default value like $t = 0.5$ generally works well in practice, suggesting that precise estimation of $p(+|x)$ is often unnecessary.
>
>
>
> 3.  **Scenarios for $g_\theta$ misspecification and mitigation:**
>     The reviewer asked for intuitive examples where the learned density ratio model $g_\theta$ might be misspecified, even when initialized from the reference policy. Misspecification can occur if the true preferred distribution $p^+(y|x)$ is too complex to be accurately represented by the chosen model architecture, given its parameterization (e.g., the LLM has insufficient capacity). In that case, the model might not be able to capture the true function even with infinite data. The bias term in our analysis (Section 3.2) relates directly to this discrepancy between the best possible model within the chosen hypothesis space and the true function; it depends on the model class, not the data distribution. Strategies to mitigate this bias primarily involve using model architectures known for their expressive power and ensuring the model has sufficient capacity (e.g., using larger models or architectures proven effective). Therefore, processing data (including prompt selection strategies) do not reduce the model bias, although they influence the data distribution which affects the variance term.
>
>
>
> 4.  **Inconsistency between Figure 1 and Section 5.2:**
>     Thank you for pointing out this discrepancy. As you mentioned, the "GSM8K" in the text describing Figure 1 should have referred to "AlpacaEval LC Winrate". We apologize for this mistake and will correct it in the camera-ready version.
>
>
>
> 5.  **DDRO underperformance on AlpacaEval:**
>     Following the correction above, the reviewer asked for insights into why DDRO might specifically underperform KTO on AlpacaEval LC Winrate. We hypothesize that KTO's specific preference model assumptions might align particularly well with the nature of the AlpacaEval benchmark. This could lead to strong performance on this specific benchmark, potentially at the expense of broader generalization. DDRO, lacking such model-specific biases, results in strong results across a wider array of benchmarks.
>
>
>
> 6.  **Error in Equation (6) and MMLU reference:**
>     Thank you for identifying the potential error in the KL term in Equation (6) and the broken MMLU reference in Section 5.1. We will carefully review Equation (6) to ensure correctness regarding the preferred/unpreferred KL terms and fix the MMLU citation. We will perform a thorough proofread to catch any similar issues.
>
>
> We appreciate the reviewer's constructive feedback, which will help enhance the clarity and correctness of our paper.

---

> > ### Comment · Reviewer_v1jw · 2025-04-04
> >
> > Thank you for your response and clarifications. I will maintain my score.

---

### Official Review · Reviewer_oR36 · 2025-03-17

**Overall Recommendation:** 3

**Summary:**

This paper focuses on LLM alignment, aiming to address two key limitations of existing direct preference learning methods such as DPO: (1) Existing work requires a preference model (e.g., the Bradley-Terry model), which may not generalize well to capture complex human preferences that do not fit these models; (2) Existing work lacks statistical consistency guarantees, indicating they do not ensure convergence to an optimal model. To handle these challenges, the authors propose Direct Density Ratio Optimization (DDRO), demonstrating that the ground-truth reward can be represented as the density ratio between positive and negative response distributions. They develop a learning approach based on Bregman divergence to estimate this ratio. Theoretically, they prove that DDRO guarantees statistical consistency, and empirically, they demonstrate that DDRO performs on par with or better than existing baselines.

**Claims And Evidence:**

This paper makes two key claims:

1. Existing SFT methods lack statistical consistency guarantees, indicating that even with sufficient training data, they do not necessarily converge to the optimal model. The authors provide a theoretical proof in Sec. 4.1 and partially justify it empirically in Sec. 5, where DDRO achieves comparable (or slightly better) results.

2. Current methods rely on predefined preference models, such as the Bradley-Terry or Plackett-Luce models, making them unsuitable for capturing more complex human preferences. The authors derive DDRO theoretically in Sec. 3, demonstrating that it does not require such an explicit preference model.

However, while both claims are reasonable and well-motivated, they are not well-supported empirically.
* For Claim 1, although DDRO does not require an explicit preference model, could the density ratio g = p- / p+ itself be regarded an implicit preference model? Besides, in Fig. 1, DDRO—which is theoretically statistically consistent—does not achieve a significant advantage over other unpaired baselines, such as KTO and BCO. According to line 27 (right part), statistical consistency is defined as "as the amount of data increases, the learned model converges to the true optimal model." A straightforward way to validate this would be to train DDRO and baselines with varying dataset sizes and compare its convergence speed to KTO, demonstrating both that statistical inconsistency is an issue in practice and that the proposed method effectively addresses it.
* For Claim 2, the authors state in line 24 (right part) that "This assumption may not accurately capture the complexity of real human preferences," which is intuitively plausible but not well justified. Experimentally, preference-model-based methods like KTO and DPO do not perform significantly worse than DDRO. To support this claim, the authors should evaluate their method on datasets with more complex human preferences and empirically demonstrate the benefits of not relying on a predefined preference model.

Generally, I believe this paper makes sufficient theoretical contributions, and the use of density ratio is also interesting. I will raise my score if the authors can address the concerns above.

**Essential References Not Discussed:**

There are already some ratio-like loss papers. The authors should cite and discuss them. Specifically:

* Zeng et al., Token-level Direct Preference Optimization. 2024.
* Hong et al., ORPO: Monolithic Preference Optimization without Reference Model. 2024.
* Xu et al., Contrastive Preference Optimization: Pushing the Boundaries of LLM Performance in Machine Translation. 2024.

**Experimental Designs Or Analyses:**

The experimental design in this paper does not fully support its claims (see [Claims and Evidence]). Additionally, several existing density ratio-like LLM alignment methods have not been cited. The authors should cite these works and discuss how DDRO compares to them. Ideally, they should include the comparison with at least one of these related methods. For relevant papers, see [Essential References Not Discussed].

**Methods And Evaluation Criteria:**

As discussed above, the two claims are well-motivated and intuitively reasonable through the proposed method, with the authors providing theoretical proofs for some aspects. However, since this work is not purely theoretical, additional experimental results are needed to empirically validate these claims. See [Claims and Evidence].

**Other Comments Or Suggestions:**

1. Line 165, left part, he -> the

2. Line 357, right part, reference missing in MMLU

**Other Strengths And Weaknesses:**

N/A

**Questions For Authors:**

1. In the unpaired ltrafeedback-gpt-3.5-turbo-helpfulness dataset, how are the unpaired negative samples used in Eq. (6) generated?

2. In Fig. 1, DDRO outperforms KTO in AlpacaEval LC Winrate. How should this result be explained?

3. In Eq. (6), is there a possibility that some positive samples have almost zero probability mass in the LLM’s space? If so, how can this issue be mitigated?

**Relation To Broader Scientific Literature:**

As far as I know, the two claims made in this paper have not been explicitly discussed in other LLM alignment research. However, several papers have already explored unifying different DPO variants within a unified framework, making this work closely related to those efforts. Specifically:

* Tang et al., Generalized Preference Optimization: A Unified Approach to Offline Alignment. 2024.
* Han et al., f-PO: Generalizing Preference Optimization with $f$-divergence Minimization. 2024.

**Theoretical Claims:**

The authors present theoretical claims in Sec. 3, Proposition 3.3, and Theorem 4.1. However, I could not find the proof for Proposition 3.3. I reviewed the derivations in Sec. 3 and did not notice obvious issues. I also briefly examined the proof of Theorem 4.1 and did not identify clear errors. However, as I am not an expert in learning theory, I cannot guarantee the correctness of the proofs.

---

> ### Author Rebuttal · Authors · 2025-04-01
>
> Thank you for your thoughtful review and constructive comments on our paper. We appreciate the opportunity to address your concerns and clarify aspects of our work.
>
>
> 1.  **Is the density ratio $g$ an implicit preference model?**
>
>     We thank the reviewer for this insightful question. The density ratio $g = p^- / p^+$ reflects inherent preferences derived directly from the true distributions ($p^+, p^-$) without assuming a specific parametric structure (like the Bradley-Terry model used implicitly by methods like DPO or KTO). Therefore, $g$ represents the data's preference information itself, not a constructed model, allowing DDRO to avoid explicit preference modeling.
>
>
> 2.  **Performance comparison and statistical consistency:**
>      We appreciate your suggestion. However, our primary theoretical claim regarding statistical consistency focuses on the convergence target – ensuring the learned model converges to the true optimal distribution under certain conditions – rather than the rate of convergence. Our experiments aim to show DDRO achieves competitive or superior performance on various benchmarks at a fixed data size, implicitly supporting the benefit of a sound convergence target.
>
>
>
> 3.  **Justification for avoiding explicit preference models:**
>     We note that the very fact that our method demonstrates advantages on the majority of benchmarks, despite underperforming on a single benchmark, suggests that preference-modeling-methods may have become overly specialized to the particular types of preferences required only by certain benchmarks.
>     Also, we believe that demonstrating the limitation of specific models (like the Bradley-Terry model assumed in DPO) does not necessarily require datasets with highly complex preferences. It is sufficient to show scenarios where preferences, even simple ones, deviate from the assumed model structure.
>     For example, Bradley-Terry model does not capture a non-transitive relatoinship such as in rock-paper-scissors.
>     We acknowledge that including demonstration of such cases would strengthen our argument and therefore plan to add a discussion or a simple illustrative example to the camera-ready version.
>
>      -  [1] Pan, Y., Tsang, I. W., Chen, W., Niu, G., \& Sugiyama, M. (2022). Fast and robust rank aggregation against model misspecification. Journal of Machine Learning Research, 23(23), 1-35.
>
> 4.  **Proof for Proposition 3.3:**
>     We apologize for the omission. We had intended the paragraph preceding Proposition 3.3 to serve as an justification. We will provide a formal proof for Proposition 3.3 in the appendix of the camera-ready version.
>
> 5.  **Missing citations for related work:**
>     Thank you for bringing these relevant papers to our attention.     We have reviewed them carefully.
>     We acknowledge these papers implicitly involve density-ratio through regularization (like odds ratios) or KL constraints.
>     However, these are primarily for stabilization or performance enhancement, and they do not appear to directly estimate or optimize the density ratio itself as the main target.
>     In contrast, the central contribution of our paper is the introduction of a consistent theoretical framework built directly upon the perspective of direct density ratio matching.
>     Therefore, while valuable contributions, we believe their direct relevance to our paper's central novelty is limited.
>
>
> 6.  **Typos and missing references:**
>     We will correct these errors and conduct a thorough proofreading using automated tools and careful manual checks to identify and fix any other potential issues in the final version.
>
>
>
> 7.  **Generation of negative samples in unpaired Ultrafeedback:**
>     Unpaired datasets like Ultrafeedback typically consist of triplets: a prompt, a generated response, and a binary label (e.g., "good" or "bad") assigned by human annotators. Therefore, responses labeled as "bad" constitute the set $D^-$.
>
> 8.  **DDRO vs. KTO on AlpacaEval LC Winrate:**
>     We interpret this result as potentially stemming from KTO's underlying preference model (based on spectrum theory) aligning particularly well with AlpacaEval benchmark. This focused alignment might come at the cost of performance on other, broader benchmarks. In contrast, DDRO, by directly learning from the preference signals without an intermediate model, avoids such model-induced biases, leading to more robust and generally strong performance across a wider range of benchmarks, as observed in our experiments (e.g., BBH, GSM8K, MMLU, TruthfulQA).
>
>
>
> 9.  **Potential for zero probability mass in Eq. (6):**
>     As mentioned in the paragraph just before Eq. (6), this can indeed occur and potentially lead to gradient spikes. To mitigate this, we introduced the function $S(\cdot)$ which shapes the loss landscape to be more benign.
>
>
> Thank you again for your detailed feedback, which will help us improve the final version of our paper.

---

> > ### Comment · Reviewer_oR36 · 2025-04-09
> >
> > Thank the authors for the responses. I think my minor concerns (e.g., the proof of Proposition 3.3) have been addressed, but the major two, (1) empirical benefits of statistical consistency and (2) justification of Claim 2, still remain.
> >
> > I fully acknowledge this work's theoretical contribution in ensuring convergence to a sound target— this is why I'm leaning toward accept. However, as mentioned before, the authors didn't frame the paper as purely theoretical work, and hence empirical evidence is important. It’s still unclear whether such theoretical benefits would translate into practical improvement. Regarding my concern on Claim 2, the non-transitive preference example is somewhat helpful but not fully convincing, as the influence of non-transitivity on performance remains uncertain.
> >
> > Therefore, I maintain my current score, though I would raise it to 3.5 if there is such an option.

---

### Decision · Program_Chairs · 2025-05-01

**Decision:**

Accept (poster)

**Comment:**

Reviewers broadly agree the paper is clear, with sound theory and empirical evaluations.

I think the paper could have engaged more deeply with the large body of related work on optimizing using preference labels directly, and the impact would be improved by giving this some attention prior to publication.